# Postbiotic of *Pediococcus acidilactici* GQ01, a Novel Probiotic Strain Isolated from Natural Fermented Wolfberry, Attenuates Hyperuricaemia in Mice through Modulating Uric Acid Metabolism and Gut Microbiota

**DOI:** 10.3390/foods13060923

**Published:** 2024-03-18

**Authors:** Lu Ren, Shangshang Wang, Shiting Liu, Hetti Arachchige Chalani Prasanthi, Yuechan Li, Jun Cao, Feiliang Zhong, Le Guo, Fuping Lu, Xuegang Luo

**Affiliations:** 1Key Laboratory of Industrial Fermentation Microbiology of the Ministry of Education and Tianjin Key Laboratory of Industrial Microbiology, College of Biotechnology, Tianjin University of Science and Technology, Tianjin 300457, Chinalfp@tust.edu.cn (F.L.); 2Ningxia Key Laboratory of Clinical and Pathogenic Microbiology, School of Laboratory Medicine, Ningxia Medical University, Yinchuan 750004, China

**Keywords:** hyperuricaemia, *Pediococcus acidilactici* GQ01, postbiotics, uric acid, intestinal microorganisms

## Abstract

Hyperuricaemia (HUA) is a disorder of purine metabolism, which manifests itself as an increase in uric acid production and a decrease in uric acid excretion, as well as a change in the structure of the intestinal microbiota. Most of the drugs currently used to treat HUA have significant side effects, and it is essential to find a treatment for HUA that is free of side effects. In this study, a novel strain, *Pediococcus acidilactici* GQ01, was screened from natural fermented wolfberry. The effects of both live bacteria GQ01 and its heat-killed G1PB postbiotic on HUA were investigated. The results showed that both probiotic GQ01 and G1PB postbiotics could effectively decrease blood uric acid, creatinine, and urea nitrogen levels in the HUA mice model. *P. acidilactici* GQ01 was more effective in inhibiting ADA activity, while G1PB postbiotics was more effective in inhibiting XOD activity. Meanwhile, GQ01 and G1PB were able to ameliorate liver and kidney tissue injury, upregulate the expression of ABCG2 in kidney and XOD gene in liver, downregulate the protein expression of URAT1 and GLUT9 in kidney, and therefore reduce the value of blood uric acid by decreasing the uric acid reabsorption and increasing the excretion of uric acid. Additionally, both probiotics and postbiotics could regulate the intestinal microbiota structure of HUA mice, so as to bring the dysfunctional intestinal composition back to normal. Furthermore, *P. acidilactici* GQ01 and G1PB postbiotics can increase the levels of acetic acid, propionic acid, and butyric acid in the intestinal tract, improve the intestinal function, and maintain the healthy homeostatic state of the intestinal tract. In summary, *P. acidilactici* GQ01 and its G1PB postbiotics may be developed as functional food or drug materials capable of treating HUA.

## 1. Introduction

Hyperuricaemia (HUA) is when the level of uric acid in the blood is higher than the solubility limit (around 6.8 mg/L) [1]. It is a common biochemical issue that shows the extracellular fluid is oversaturated, often due to too much uric acid being produced or not enough being removed by the kidneys. This imbalance in uric acid levels increases the likelihood of developing gout in individuals with HUA [2]. Generally, male patients are more likely than female patients, and the age of patients tends to be younger [3,4]. HUA is also accompanied by complications such as gout [5], chronic kidney disease [6], obesity [7], diabetes [8], and hypertension [9]. HUA is a significant risk to individuals’ well-being and health and increases the public health burden on society. Modern medicine believes that HUA is mainly linked to consuming a diet rich in purine, fructose, and protein, which triggers purine metabolism disorders, resulting in disorders and imbalance of the purine metabolism end product, uric acid, with accumulation in the blood and elevated levels that can lead to a series of adverse reactions and kidney and other organ damage. Often, dietary approaches do not succeed in adequately managing the levels of uric acid in the body. Drug therapy for hyperuricaemia is hindered by the toxic side effects and high cost of commercially available medications, highlighting the necessity for more affordable and safer treatment options in clinical practice.

A significant link between HUA and changes in gut flora has been suggested, as shown by recent studies. HUA affects intestinal epithelial integrity and gut microbial homeostasis [5]. Intestinal microbiota and its metabolites play an important role in human uric acid metabolism; intestinal flora can participate in purine absorption, synthesis of uricase, and affect the intestinal mucosal barrier, and thus are directly or indirectly involved in uric acid production and excretion [10]. A persistently high uric acid level induces chronic inflammation in the gut, with consequent changes in the species, number, and distribution of the bacterial flora [11]. Alterations in the gut microbiota can worsen chronic inflammation in the intestines, leading to damage to the intestinal lining, changes in the permeability of the intestines, and hindering the normal absorption and transportation of uric acid, ultimately causing an increase in blood uric acid levels [12]. Additionally, the gut microbiota plays a role in purine metabolism through its proxies. Intestinal bacteria also play a role in purine breakdown by producing metabolites that impact the activity of proteins responsible for transporting uric acid in the intestines, ultimately controlling the levels of uric acid in the blood [13]. Furthermore, intestinal bacteria have the ability to convert UA into very soluble allantoin and can also generate SCFAs to control UA metabolism. Consequently, the presence or lack of UA imbalance in the gut microbiota could impact UA elimination, resulting in elevated levels of serum UA and ultimately causing hyperuricaemia [14,15].

Probiotics in the intestinal flora are beneficial to the human body. According to the definition of the Joint Expert Committee of the Food and Agriculture Organization of the United Nations (FAO) and the World Health Organization (WHO), probiotics are live microorganisms that, when ingested in sufficient quantities, have a beneficial effect on the health of the host [16]. Probiotics are able to colonise the human intestinal and reproductive tracts, where they play a key role in regulating lipid, glucose, and uric acid metabolism [17]. Probiotics usually include *Lactobacillus*, *Bifidobacterium*, *Plicococcus*, etc. [18]. In recent years, inactivated microorganisms (metabolites) have been found to be the material basis for their action. In 2012, people first proposed the concept of postbiotics [19], which are defined as inactivated microorganisms and/or microbial constituents that are beneficial to the health of the host, including metabolites produced during fermentation, active small molecules released from decomposition of the fermentation substrate, and dead and lysed cellular fractions. In May 2021, the International Society for the Science of Probiotics and Prebiotics (ISAPP) released a consensus statement on postbiotics, making postbiotics a global category of interest in the field of intestinal microecology and defining postbiotics as “preparations of inanimate microorganisms and/or their constituents that have a beneficial effect on the health of the host” [20]. Postbiotics, in comparison to their live bacterial counterparts (probiotics), demonstrate enhanced stability, extended shelf life, ease of storage and transport, and greater safety profiles, particularly for sensitive populations where probiotic live cultures may pose potential risks [21,22]. Additionally, postbiotics are usually easier to be absorbed by the intestinal tract, improving their utilisation [23,24].

Wolfberry (*Lycium barbarum* L.) belongs to the Solanaceae family of plants; with a variety of health benefits, it not only can be processed into food but is also used to nourish the liver and kidneys in traditional Chinese medicine. It has a long history of use as medicine and food [25]. Previous studies have shown that wolfberry can reduce blood uric acid levels and regulate the balance of intestinal microflora in HUA mice [8,26]. Probiotic fermented fruits, vegetables, and dairy products exist in large quantities on the market [27,28]. Fermented products not only have the original taste and nutrients, but also produce the unique flavour and nutrients of fermentation [29,30]. So, it is important to screen probiotics with good fermentation properties. In this study, we screened probiotics with blood uric acid lowering function from natural fermented wolfberry (Jiaosu) for the first time and investigated the effects and mechanisms of the uric acid lowering properties of live probiotics and their postbiotics.

## 2. Materials and Methods

### 2.1. Reagents and Materials

Potassium oxonate (PO, with a minimum purity of 98%) and allopurinol were bought from Sigma-Aldrich (St. Louis, MO, USA). Yeast extract was purchased from Sigma-Aldrich (St. Louis, MO, USA). Solarbio Science and Technology Co., Ltd. (Beijing, China) supplied carboxymethyl cellulose sodium (CMC), while Nanjing JianCheng Bioengineering Institute (Nanjing, China) provided assay kits for blood uric acid (UA), xanthine oxidase (XOD), blood urea nitrogen (BUN), and creatinine (CR). Thermo Fisher Scientific (Waltham, MA, USA) supplied Trizol from Invitrogen (Waltham, MA, USA).

### 2.2. Bacterial Strain Isolation, Identification, and Culture

The dried wolfberry and sterile water were packed in sterile bottles at a mass ratio of 1:5, placed in a cool place, and stored in an enclosed place for 35 days. Take 1 mL of natural fermented wolfberry and apply it to the solid selection medium of lactic acid bacteria with gradient dilution, pick the single colony with an obvious calcium solubilising circle, and purify and isolate to obtain lactic acid bacteria. The genome of the obtained single strain was extracted using a kit (TIANGEN, Beijing, China), and PCR amplification was performed using random primers 27F (AGAGTTTGATCMTGGCTCAG) and 1492R (GGTTACCTTGTTACGACTT), and finally, the PCR products were sent to GENEWIZ Biotechnology Co. (Tianjin, China). *P. acidilactici* GQ01 was isolated from natural fermented wolfberry and conserved in the General Microbiology Centre of China Microbial Strain Conservation and Management Committee (CGMCC) with the strain conservation number CGMCC NO. 21609. A total of 2% (*v*/*v*) GQ01 was inoculated in an MRS liquid medium and incubated at 37 °C; the organisms after centrifugation were washed twice with saline, and G1PB was prepared via constant heating at 65 °C for 30 min.

### 2.3. Determination of Growth Kinetic and Organic Acids Production

*P. acidilactici* GQ01 strains (2%) were inoculated in liquid MRS (Difco, Detroit, MI, USA), cultured at 37 °C, and the number of bacteria in the bacterial suspension per unit volume was measured at OD_600nm_ with simultaneous recording of pH in MRS liquid medium.

### 2.4. Animal Treatment

The Animal Care and Ethics Committee of Tianjin University of Science and Technology approved the experimental procedures. All handling of animals was conducted following the guidelines in the Manual for the Treatment and Utilization of Laboratory Creatures. Fifty male Kunming mice, 4 weeks old, free of pathogens, and weighing between 18 and 22 g, were acquired from SPF Biotechnology Co., Ltd. (Beijing, China), with Quality Certificate SCXK (JING) 2019-0010. Prior to this study, the test subjects were given time to adjust and were kept in a controlled environment for 7 days, following a 12 h light/dark schedule, with a room temperature maintained at 24 ± 2 °C and humidity levels at 52 ± 5%.

Fifty Kunming breed mice were divided into five groups, each group with 10 mice. This study included a control group (NC), a group with a HUA model (M), a group treated with the positive drug allopurinol (AP), a group receiving intervention with live bacteria GQ01 (GQ01), and a group receiving intervention with postbiotics from GQ01 (G1PB). Formal experiments began following one week of mice being fed adaptively. Each day, the mice were weighed and given doses based on their weight. All mice except those in the NC group were gavaged with 10 g/kg of yeast paste and 300 mg/kg of potassium oxonate intraperitoneally. One hour later, the GQ01 group and G1PB mice were gavaged with 10^8^ cfu/mL of GQ01 and inactivated GQ01 bacterial suspension resuspended in saline, and the AP group was gavaged with 30 mg/kg of allopurinol. The NC and M groups were gavaged with the corresponding volume of stroke physiological saline solution (SPSS). The whole experiment lasted for 21 days. Blood samples were obtained by removing the eyeballs 1 h after the last dose. Afterwards, mice were killed via cervical dislocation, and their livers, kidneys, small intestines, colon contents, and cecum contents were dissected. The liver, kidney, small intestine, colon, and cecum were stored at −80 °C.

### 2.5. Biochemical and Histopathological Analyses

Following euthanasia, venous blood was obtained from mice and the liquid portion was spun at a slow rate before being frozen at −80 °C. The serum was tested for UA, BUN, CR, ADA, and XOD activity using a Nanjing JianCheng Bioengineering Institute kit. Body weight changes were monitored every five days until the cycle’s completion. Each mouse was weighed prior to euthanasia, and the liver and kidney weights were measured post-mortem to calculate the organ index using a specific formula.

Liver and kidney tissues soaked in 4% paraformaldehyde were cleaned, fixed in 10% formalin, embedded in paraffin, and cut into 4 μm thick sections. Following deparaffinization and rehydration, the samples underwent staining with hematoxylin and eosin (H&E). An orthogonal fluorescence microscope (Olympus BX53, Tokyo, Japan) was used to observe the slides in a blinded manner. At least 3 samples per group and from each section were randomly selected for analysis at 200× magnification.

### 2.6. RT-qPCR Analysis

Liver and kidney tissues from mice were used to extract total RNAs with Trizol reagent (Takara, Kyoto, Japan). A total of 2 mg of RNA was converted to cDNA using LunaScript RT SuperMix Kit (NEB, Ipswich, MA, USA) as per the manufacturer’s guidelines. Gene expression analysis was performed by conducting RT-PCR with the SYBR Green Premix Pro Taq HS qPCR kit from Accurate Biotechnology in China. All gene expression levels were normalized to the levels of β-actin. Table 1 displays the primers that were utilized. All primers used in this study were synthesised by GENEWIZ Biotechnology Co. (Tianjin, China). The relative mRNA levels of the PCR products were quantified using β-actin as an internal standard and calculated with the 2^−ΔΔCt^ method.

### 2.7. Western Blot Analysis

After washes with PBS three times, a 10-fold volume of RIPA lysis buffer was added to the kidney tissue for homogenisation. Afterwards, the homogenate was mixed with the protease inhibitor PMSF (1 mM) while kept cold. Then, 12% sodium dodecyl sulphate-polyacrylamide gel electrophoresis (SDS-PAGE) was used to separate ten micrograms of sample, with electrophoresis conducted at 150 V for 60 min. The proteins that were fully isolated were subsequently moved to a PVDF transfer membrane using a current of 500 mA. The membrane was closed for two hours with 10% skimmed milk fully dissolved in TBST. The PVDF transfer membrane was placed in the appropriate antibody incubator, where 10 mL of the corresponding primary antibody was added and kept away from light for 2 h. Following a 30 min TBST wash, 10 mL of the corresponding secondary antibody was added for 1 h. The chemiluminescence signals of the specific target proteins were captured and measured using an infrared laser imager (LI-COR, Lincoln, NE, USA) in comparison to β-actin; after another 30 min TBST wash, chemiluminescent signals were photographed.

### 2.8. Determination of SCFAs via GC

Detection of SCFAs in the mouse colon contents was performed using gas chromatography (Agilent GC7890A, Santa Clara, CA, USA). The instrumental parameters included an injection volume of 1 µL, an inlet temperature of 250 °C, a split ratio of 1:1, a chromatographic column of HP-5 (19091J-413) with dimensions of 30 cm length, 320 µm inner diameter, and 0.25 µm film thickness, and a column flow rate of 2 mL/min. The starting temperature of the column is 90 °C, which is maintained for 6 min before being increased to 200 °C at a rate of 10 °C per minute. The heater is set to 250 °C, with an H_2_ flow rate of 30 mL/min and an air flow rate of 300 mL/min.

### 2.9. Gut Microbiota Analysis

The contents of mouse cecum were delivered to Majorbio Technology Co. (Shanghai, China) for 16S rDNA sequencing. The detailed procedure involved grinding and crushing the bacteria, extracting DNA of intestinal microbiota using a DNA extraction kit, and amplifying the V3–V4 region of 16S rDNA with specific primers 338F and 806R (338F sequence: 5′-ACTCCTACGGGGAGGCAGCAG-3′, 806R sequence: 5′-GGACTACHVGGGTWTCTAAT-3′). The UPARSE pipeline was used to group valid tags into operational taxonomic units (OTUs) with a similarity of at least 97%. The RDP classifier, utilizing a basic Bayesian model, was employed to assign representative sequences to different organisms based on the SILVA database.

### 2.10. Statistical Analysis

Quantitative information was presented as the average plus the standard error of the average (SEM). SPSS Statistics 23.0 was utilized to analyse the data through a one-way analysis of variance followed by Duncan’s post hoc test. GraphPad Prism 8.0 software was used to create the drawings.

## 3. Results and Discussion

### 3.1. Strain Identification and Biochemical Properties

The colonies appeared globular, Gram-positive, partially anaerobic, and small and white on the MRS medium (Figure 1A). The results of colony amplification sequencing were compared with the NCBI BLAST database, and GQ01 was closely related to *P. acidilactici*, which is a Gram-positive bacterium that has been included in China’s List of Bacterial Strains Available for Use in Food in 2001 (Figure 1B). From Figure 1C, it can be seen that GQ01 grows slowly during the initial 4 h of incubation, enters the logarithmic growth period after 6 h of incubation, and the value of OD_600nm_ levels off at 12 h of incubation and then enters the stabilisation period, with the OD_600nm_ being 2.58 and remaining essentially unchanged after 30 h. The rate of acid production is one of the important characteristics of good viability of *lactic acid bacteria* and is also an important indicator for screening good strains. Acid-producing capacity is an important characteristic of *lactic acid bacteria* [31]. There is a correspondent relationship between the trend of growth and the trend of acid production (Figure 1D). For 0~2 h the delayed period is evident in which the concentration of the bacterium did not increase significantly and the pH changed little. After 2 h into the logarithmic growth period, the bacteria growth and metabolism results in rapid consumption of carbon sources, producing a large number of organic acids with the pH decreasing rapidly. After 12 h growth reaches the stable period and the concentration of bacteria gradually tends to stabilise; pH also gradually tends to stabilise with a final pH value of 3.98–4.01.

### 3.2. P. acidilactici GQ01 and G1PB Postbiotics Maintain Body Weight and Reduce Blood Uric Acid, Blood Creatinine, and Blood Urea Nitrogen in HUA Mice

Under normal conditions, the ratio of organ weight to body weight is relatively constant. When an animal is contaminated, the weight of the damaged organs changes, and therefore the organ coefficients change [32]. An increase in the organ coefficient indicates congestion, oedema, or hyperplasia of the organ, whereas a decrease in the organ coefficient indicates atrophy and other degenerative changes in the organ; so, the organ coefficient can be used to assess whether the organ is damaged or not [12,33]. The results showed that the body weights (Figure 2A) of the mice were all on the rise, with the model group growing faster than other groups, the allopurinol group growing slowly, and the body weights of the mice declining with the reduction in diet in the course of the experiment. There was no significant difference in body weight between the groups, but the body weight of the model group was significantly higher than that of the normal group, while that of the allopurinol group was significantly lower than that of the normal group. Compared with the normal group, the liver coefficient (Figure 2B) of the model group was significantly higher (*p* < 0.001), and it was found to be enlarged during autopsy, whereas the liver coefficients of the GQ01 group (decreased by 4.68%) and the G1PB group (decreased by 9.27%, *p* < 0.05) tended to be similar to those of the normal group. The kidney coefficients (Figure 2C) of the model group were elevated (increased by 11.47%), whereas those of the allopurinol group were significantly reduced (reduced by 11.18%, *p* < 0.001), and the kidneys of the model group were found to be enlarged during autopsy, and the renal coefficients of the G1PB group (1.35% ± 0.18%) tended to be the same as those of the normal group (1.40% ± 0.10%).

As shown by the results, compared with the normal group, the serum uric acid of mice in the model group was significantly elevated by 43.20% (*p* < 0.001), indicating that the HUA model was successfully established (Figure 2D). Compared with the model group, the uric acid levels in both the GQ01 and G1PB groups were significantly reduced by 40.87% and 35.18% (*p* < 0.001). In order to verify the protective effects of GQ01 and G1PB on the liver and kidney of HUA mice, CRE and BUN levels were measured. The results showed that, compared with the normal group, the CRE and BUN levels in the model group were significantly increased by 50.39% and 53.56% (*p* < 0.001), and the BUN and CRE levels in the allopurinol group (increased by 51.72%) were higher than those in the normal group, with the BUN level being much higher than those in the normal group (increased by 10.19%) and the model group (increased by 48.51%) (*p* < 0.001) (Figure 2E,F). This verified the previously reported phenomenon of adverse effects of allopurinol. With the treatment of GQ01 and G1PB, the levels of CRE were reduced by 24.72% and 44.46%, *p* < 0.01), and BUN levels were reduced by 54.22% (*p* < 0.001) and 36.98% (*p* < 0.01). Respectively, relative to the model group, the levels of the indexes tended to converge with those of the normal group, which indicated that *P. acidilactici* GQ01 and G1PB postbiotics were able to alleviate the renal injury caused by HUA.

### 3.3. P. acidilactici GQ01 and G1PB Postbiotics Reduces Mice Blood UA and Inhibits ADA and XOD Activity

The formation of uric acid in the body is associated with two enzymes, ADA and XOD, which are the key enzymes that catalyse and regulate the production of uric acid, of which ADA catalyses the conversion of adenosine to produce inosine [34], which further produces hypoxanthine and xanthine, indirectly catalysing the formation of uric acid, whereas XOD directly catalyses the gradual oxidation of hypoxanthine and xanthine to form uric acid and also converts the protein in food to purine to eventually form uric acid [2,17,35].

The results show that inhibition of ADA activity can reduce uric acid production (Figure 3A), thus reducing uric acid levels in serum. Compared with ADA levels in the normal group (2.36 ± 0.64 U/L), ADA levels in the model group were significantly increased (*p* < 0.0001) to 5.31 ± 0.98 U/L. The positive control group was administered allopurinol, which functions as a uric acid lowering drug and did not play a role in reducing ADA viability, and the enzyme activity level (6.98 ± 0.98 U/L) was significantly higher than that of the normal group (*p* < 0.0001) and the model group (*p* < 0.001). The treatment groups, the GQ01 and G1PB groups, demonstrated a significant reduction in ADA viability to 2.58 ± 0.45 U/L (*p* < 0.0001) and 4.58 U/L (*p* < 0.0001) and 4.48 U/L (*p* < 0.0001), with the ADA viability of the GQ01 group converging with that of the normal group, with 4.06 ± 0.61 U/L (*p* < 0.001). Elevated XOD activity usually leads to overproduction of uric acid, which in turn leads to HUA (Figure 3B), with XOD viability in the model group (15.97 ± 1.28 U/L) being higher than that of the normal group (9.74 ± 0.58 U/L) with the increase reaching statistical significance (*p* < 0.0001). The positive control allopurinol group exhibited significantly decreased XOD enzyme activity (7.91 ± 1.04 U/L) compared to the model group (*p* < 0.0001). The treatment groups GQ01 and G1PB also exhibited significantly reduced XOD activity with levels of 12.63 ± 1.19 U/L and 10.28 ± 0.54 U/L. Xanthine oxidase activity in serum and XOD mRNA expression in the liver were significantly enhanced in the model group (Figure 3C). Allopurinol as an XOD inhibitor also showed downregulation of XOD gene expression, and both the GQ01 and G1PB group exhibited downregulated expression of XOD and the GLUT9 gene, a relevant transporter protein for evaluating the abnormalities of uric acid excretion in HUA, which has been proved to be altered in patients with gout and HUA. XOD enzyme activity in the G1PB group tended to be the same as that of the normal group. Therefore, inhibition of ADA and XOD may be the underlying mechanism of the uric acid-lowering function of GQ01 and G1PB, with GQ01 inhibiting ADA viability more effectively and G1PB inhibiting XOD viability more effectively, and with both of them achieving effective reduction in blood uric acid.

### 3.4. P. acidilactici GQ01 and G1PB Postbiotics Ameliorate Liver and Kidney Injury

The results of H&E staining showed that HUA modelling causes damage to the organs of mice, and the pathological features were effectively improved by treatment with GQ01 and G1PB (Figure 4). In the normal group, the hepatocytes had a rounded and compact morphology, with large, rounded, and centred nuclei, little heterochromatin and light staining, clearly visible nucleoli, and abundant hepatocyte cytoplasm. In the model group, hepatocytes were swollen and the hepatic sinusoids were slightly dilated and irregularly shaped, with inflammatory cell infiltration and fat vesicle deposition. In the allopurinol group, the hepatic sinusoids were more dilated, there was smaller hepatocyte volume and hepatocyte nuclei, and more inflammatory cells (neutrophilic infiltration) were observed in the lesions than in the model group. Glomerular atrophy and deformation, marked dilatation of tubular lumen, and mild renal oedema were observed in the model and allopurinol groups compared to the normal group. Allopurinol treatment aggravated renal oedema and tubular damage. These results were consistent with the nephrotoxicity of allopurinol. Significant improvements in these symptoms were observed in the GQ01 and G1PB groups. These findings suggest that GQ01 and G1PB may be used as probiotics and postbiotics to alleviate the symptoms of HUA and to ameliorate the hepatic and renal tissue damage.

### 3.5. P. acidilactici GQ01 and G1PB GQ01 Postbiotics Regulate the Expression of Genes Related to Uric Acid Reabsorption and Excretion

Reduced excretion of uric acid is one of the causes of HUA, in which almost 90% of uric acid is reabsorbed back into the bloodstream in the renal tubules, a process that is mainly regulated by the urate transporter protein GLUT9 [36]. It has been found that the uric acid excretory protein ABCG2 and the reabsorption protein GLUT9 are closely related to HUA [22,37]. Decreased expression of uric acid excretory proteins and increased expression of reabsorption proteins both lead to excess uric acid in the blood, ultimately leading to HUA [34]. Human URAT1 and glucose transporter protein 9 are both involved in urate reabsorption, and high levels of URAT1 and GLUT9 expression may cause HUA or gout [38,39]. In order to further investigate the mechanism of the uric-acid-lowering effects of *P. acidilactici* GQ01 and G1PB postbiotics, we examined the expression of uric-acid-excretion-related protein genes in mice and selected GLUT9 and ABCG2, which are expressed in the kidney; RT-PCR results are shown in Figure 5A,B, and the mRNA level of ABCG2 was upregulated by both GQ01, which was significant (*p* < 0.0001), and G1PB. Relative to the model group, GQ01 (*p* < 0.001) and G1PB (*p* < 0.05) significantly downregulated the expression of GLUT9. GLUT9 expression levels are abnormally elevated in vivo, and its inhibitors are used as therapeutic modalities to lower uric acid. In this study, GLUT9 gene transcription levels were significantly upregulated in mice under high-purine dietary interference, which is consistent with existing studies, and downregulated in mice after GQ01 and G1PB interference. Studies have shown that ABCG2 gene transcription was found to be downregulated in a high-uric-acid model, which is also consistent with the present study, suggesting that downregulation of GLUT9 gene transcription in kidneys, upregulation of ABCG2 in kidneys, and transcription of XOD gene in livers under the intervention of GQ01 and G1PB reduced the blood uric acid values by decreasing the uric acid reabsorption and increasing the uric acid excretion.

Furthermore, the impact of *P. acidilactici* GQ01 and G1PB postbiotics on kidney GLUT9 and URAT1 proteins were examined with Western blotting (Figure 5C). As shown in Figure 5D,E, the model group exhibited significantly upregulated GLUT9 and URAT1 expression (*p* < 0.0001) via potassium oxonate and yeast paste treatment, and GQ01 and G1PB treatment significantly downregulated GLUT9 and URAT1 protein expression (*p* < 0.0001). The data obtained suggest that GQ01 and G1PB may promote uric acid excretion by downregulating the expression of URAT1 and GLUT9 in HUA model mice.

### 3.6. P. acidilactici GQ01 and G1PB Postbiotics Regulate Gut Microbiota Dysbiosis in HUA Mice

Probiotics can exert uric-acid-lowering effects through multiple mechanisms: some probiotics are involved in the synthesis of uric-acid-degrading enzymes, including uricase and allantoinase, which help to degrade uric acid into urea, thus reducing uric acid levels, on the other hand, some probiotics have the ability of purine degradation, which can reduce serum uric acid levels and HUA by decreasing the absorption of purines in the intestinal tract [40,41,42].

In order to explore the effects of *P. acidilactici* GQ01 and G1PB postbiotics on the intestinal microbiota of HUA mice, we analysed the intestinal microbial communities of five experimental groups by sequencing the V3–V4 region of the 16S rRNA gene. Through systematic bioinformatics analysis, we found that GQ01 and G1PB restored the HUA-induced changes in the structural composition of the intestinal microbial community. Venn diagrams showed that there were 434 OTUs common to the five groups, and the composition of the OTUs was the most similar between the normal group and the GQ01 and G1PB groups (Figure 6A). Through PCoA, principal component analysis (Figure 6B), we learnt that the five experimental groups were clustered independently of each other and intersected each other, and the NC group was far away from the M and AP groups and independent of each other with no intersection, but the NC group was closer to and intersected with the GQ01 and G1PB groups. The results suggest that *P. acidilactici* GQ01 and G1PB postbiotics may regulate the gut microbiota of HUA mice to undergo structural changes and normalise the dysregulated gut composition.

The relative abundance of the five groups of samples at the phylum level is shown in Figure 6C. The dominant intestinal bacterial groups were *Bacteroidota*, *Firmicutes*, *Proteobacteria*, *Actinobacteriota*, *Patescibacteria*, *Desulfobacterota*, *Campylobacterota*, *Deferribacterota*, and *Verrucomicrobiota*. After modelling, the relative abundance of *Bacteroidota* was significantly increased in the GQ01 and G1PB groups, the relative abundance of *Firmicutes* was increased in the AP and GQ01 groups, the relative abundance of *Proteobacteria* was substantially reduced in the AP and G1PB groups, the relative abundance of *Actinobacteriota* was significantly reduced in the GQ01 and G1PB groups, the relative abundance of *Desulfobacterota* decreased in the GQ01 group, and the relative abundance of *Campylobacterota* and *Deferribacterota* increased in the G1PB group. The relative abundance of *Proteobacteria* decreased significantly in the AP and G1PB groups, and the relative abundance of *Actinobacteriota* decreased significantly in the GQ01 and G1PB groups. It can be seen that the intestinal microbiota of mice was altered under the influence of a high-purine diet, whereas GQ01 and G1PB did not change the structure of the intestinal microbiota of mice in the same way, and the allopurinol group exhibited more altered intestinal microbiota. We speculate that the way they act in the intestinal tract is also different, and more experiments are needed to verify the specific mechanism at a later stage. The relative abundance of the five groups of samples at the genus level is shown in Figure 6D. *Aerococcus* infections can cause clinical urinary tract infections, bloodstream infections, endocarditis, bone and joint infections, etc., and the relative levels of *Aerococcus* in the G1PB group were significantly different from those in the other four groups with a decrease in their relative levels, and the relative levels of *Psychrobacte* in both the GQ01 and G1PB groups were lower than in the other groups. *P. acidilactici* GQ01 increased the relative level of *Pediococcus*. *Lachnoclostridium* can exert anti-inflammatory effects and play a role in body homeostasis. *Enterococcus* is an important pathogen of infections, which not only causes urinary tract, skin, and soft tissue infections, but also causes life-threatening abdominal infections, septicaemia, cardio-periostitis, and meningitis; the relative abundance of *Enterococcus* was reduced with G1PB intervention. *Akkermansia* is a human gut bacterium that plays an important role in the renewal of the mucus layer, and its relative level was effectively restored in the GQ01 group. G1PB intervention increased the relative level of *Muribaculum*. *Corynebacterium* is a pathogenic bacterium that parasitises human or other animal bodies, and the G1PB group exhibited inhibited growth of this pathogenic bacterium, with a significant reduction in relative levels. The relationship between groups and colonies can be found in the fact that colony composition is linked between different groups (Figure 6E). With the intervention treatment with GQ01, the ratio of the Bacteroidetes phylum to the Firmicutes thick-walled phylum (Bac/Firm ratio, Figure 6F) was almost restored to the level of the NC group, indicating that GQ01 could shape the gut microbiota of HUA mice. Therefore, we conclude that in this experiment, the high-purine diet induced high-uric-acid model destroys the anaerobic environment in the intestinal tract of mice and leads to metabolic disorders and dysbiosis, and the high-purine diet alters the structure of the intestinal microbiota and clusters the growth of harmful flora, but after the intervention of *P. acidilactici* GQ01 and G1PB postbiotics, the intestinal disorders improved through the promotion of the growth of probiotics and the inhibition of the growth of some pathogenic bacteria.

### 3.7. P. acidilactici GQ01 and G1PB Postbiotics Regulate SCFAs Levels to Restore Intestinal Homeostasis

SCFAs are a group of fatty acids with fewer than six carbon atoms, which are produced by gut microorganisms fermenting undigested dietary fibres, proteins, and peptides in the intestine, and the most abundant ones are acetic acid, propionic acid, and butyric acid, which account for more than 95% of the total content of SCFAs [10]. SCFAs have a variety of beneficial effects on HUA, such as inhibition of XOD activity, promotion of uric acid secretion, etc. [22]. Studies have shown that a set of conserved genes widely present in uric-acid-consuming Enterobacteriaceae encode the uric acid degradation pathway, converting uric acid to SCFAs [43]. There have also been studies that have shown that propionic acid and butyric acid [44] are among the SCFAs that provide ATP to the cells of the intestinal wall, which excretes uric acid [45]. Acetic acid sodium and butyrate sodium reduce SUA levels in mice, and butyric acid helps maintain intestinal barrier integrity [46]. Some of the acetic and butyric acids produced in the intestine can enter the body through the bloodstream and be involved in uric acid synthesis and inhibition of XOD activity [47]. Isobutyric acid and isovaleric acid are produced via bacterial catabolism of valine and leucine [48]. The results showed that after modelling, compared to the normal group, the model group had decreased levels of acetic acid (40.74% decrease, *p* < 0.0001), propionic acid (18.51% decrease), butyric acid (34.38% decrease, *p* < 0.05), and valeric acid (51.17% decrease, *p* < 0.0001), and increased levels of isobutyric acid (12.36% increase) and isovaleric acid (27.47% increase) (Figure 7). In the allopurinol group, compared to the normal group, the intestinal levels of acetic acid (decreased by 54.34%, *p* < 0.0001), propionic acid (decreased by 44.41%, *p* < 0.0001), butyric acid (decreased by 60.86%, *p* < 0.001), and valeric acid (decreased by 56.10%, *p* < 0.0001) were significantly decreased, and isobutyric (*p* < 0.01) and isovaleric acid (*p* < 0.0001) were significantly increased. With treatment, short-chain fatty acids were significantly increased in both the GQ01 and G1PB groups, with the most significant increase in butyric acid in the GQ01 group (67.65% increase, *p* < 0.0001), which increased the total short-chain fatty acid content in the intestine. Relative to the normal group, the GQ01 and G1PB groups exhibited restored modelling-damaged short-chain fatty acid content, and the short-chain fatty acid composition more closely resembled that of the normal group.

Therefore, it is concluded that *P. acidilactici* GQ01 can increase the content of acetic acid, propionic acid, and butyric acid in the intestine, and, through the increased acetic acid, can participate in the synthesis of uric acid, can reduce the level of blood uric acid through the increase in the content of propionic acid via excretion of uric acid, and can maintain the integrity of the intestinal barriers. G1PB postbiotics can increase the content of acetic acid, propionic acid, and isovaleric acid in the intestinal tract of mice, and, as a result, treatment with G1PB can make short-chain fatty acids, and the content of SCFAs tends to be higher in G1PB-treated mice than that of the normal mouse group. The increased acetic acid participates in the synthesis of uric acid, improves intestinal function, and maintains the health of the intestinal tract.

## 4. Conclusions

In conclusion, we found that *P. acidilactici* GQ01 from natural fermented wolfberry and its G1PB postbiotics were effective in ameliorating HUA in mice. *P. acidilactici* GQ01 inhibited ADA activity and the G1PB postbiotics inhibited XOD activity, both of which reduced blood uric acid, blood creatinine, and blood urea nitrogen levels in hyperuricaemic mice. Both of them improved liver and kidney tissue damage and repaired the uric acid metabolism system in HUA mice by regulating the expression of genes and proteins related to renal reabsorption and excretion. By promoting the growth of probiotics and inhibiting the growth of certain pathogenic bacteria, *P. acidilactici* GQ01 and G1PB postbiotics restored structural changes in the intestinal microbiota of HUA mice, modulated intestinal microflora dysbiosis, and increased the levels of microbial metabolite SCFAs (e.g., acetic, propionic, and butyric acids) in the gut. These findings suggest that *P. acidilactici* GQ01 and G1PB postbiotic have great potential to be developed as functional food or pharmaceutical materials for the treatment of HUA.

## Figures and Tables

**Figure 1 foods-13-00923-f001:**
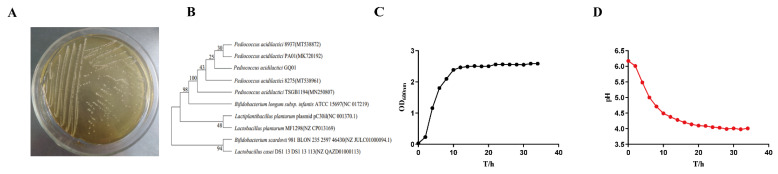
Strain identification and biochemical properties analysis of *P. acidilactici* GQ01. (**A**) Colony morphology of *P. acidilactici* GQ01; (**B**) phylogenetic tree of *P. acidilactici* GQ01 and related bacteria; (**C**) growth curve of *P. acidilactici* GQ01; and (**D**) acid production level of *P. acidilactici* GQ01.

**Figure 2 foods-13-00923-f002:**
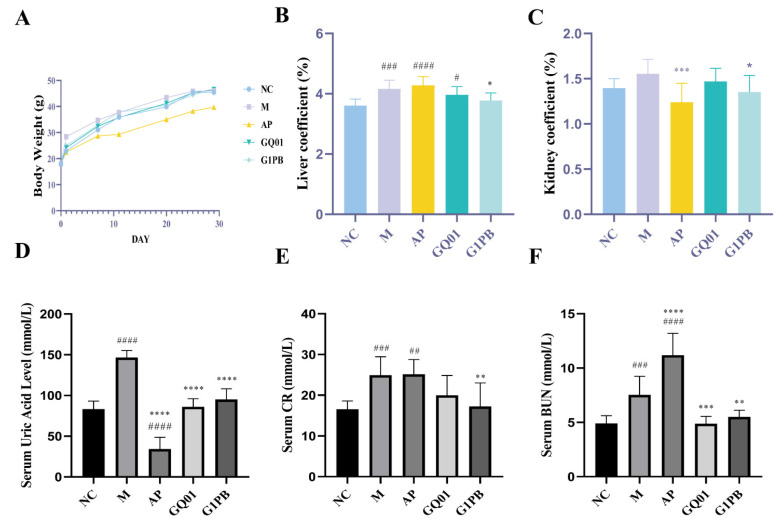
Influence of *P. acidilactici* GQ01 and G1PB on signs and physiological and biochemical indices of HUA mice. (**A**) Weight change curve; (**B**) liver index; (**C**) kidney index; (**D**) serum UA level; (**E**) serum CR level; and (**F**) serum BUN level. Data are expressed as mean ± S.E.M. (n = 10). * or ^#^ *p* < 0.05, ** or ^##^
*p* < 0.01, *** or ^###^ *p* < 0.001, **** or ^####^
*p* < 0.0001 (^#^ Compared with the NC group and * is compared to M Group). NC was control group. M was HUA model group. AP was the group treated with the positive drug allopurinol. GQ01 group was given live *Pediococcus acidilactici* GQ01. G1PB group was given inactivate *Pediococcus acidilactici* GQ01.

**Figure 3 foods-13-00923-f003:**
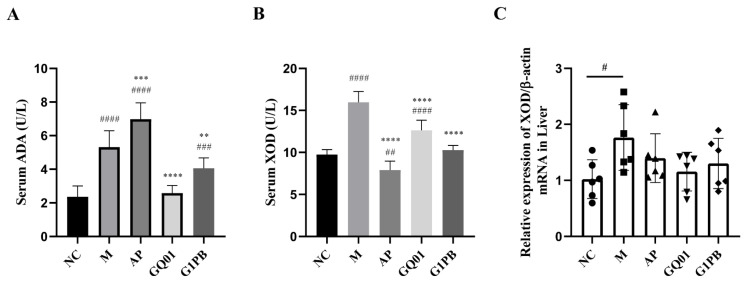
Effects of *P. acidilactici* GQ01 and G1PB postbiotics on serum ADA (**A**) and XOD (**B**), as well as the mXOD mRNA level (**C**) in liver. Data are expressed as mean ± S.E.M. (n = 10). ^#^
*p* < 0.05, ** or ^##^ *p* < 0.01, *** or ^###^
*p* < 0.001, **** or ^####^
*p* < 0.0001 (^#^ Compared with the NC group and * is compared to M Group). NC was control group. M was the HUA model group. AP group was treated with the positive drug allopurinol group. GQ01 group was given live *Pediococcus acidilactici* GQ01. G1PB group was given inactivate *Pediococcus acidilactici* GQ01.

**Figure 4 foods-13-00923-f004:**
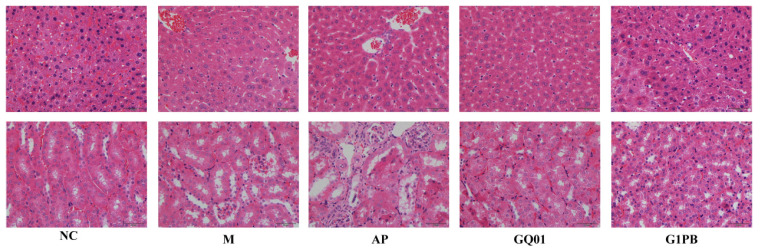
Representative micrographs of H&E-stained mouse liver tissue and kidney tissue sections (×200). NC was control group. M was HUA model group. AP group was treated with the positive drug allopurinol group. GQ01 group was given live *Pediococcus acidilactici* GQ01. G1PB group was given inactivate *Pediococcus acidilactici* GQ01.

**Figure 5 foods-13-00923-f005:**
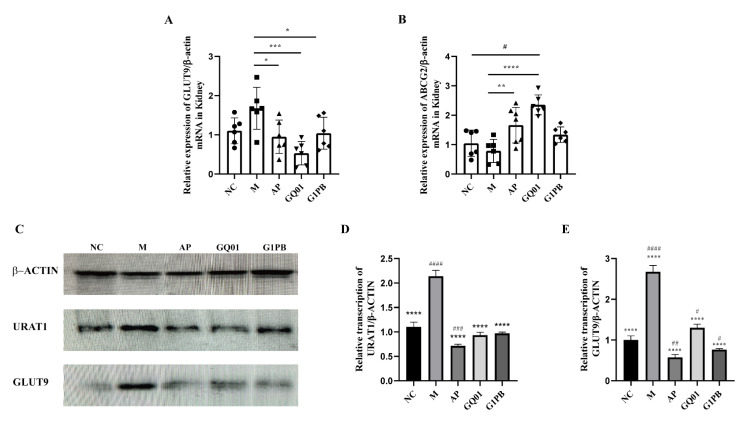
Expression of proteins associated with uric acid transport in the kidney. Gene expression levels of uric acid-related proteins in (**A**) ABCG2 and (**B**) GLUT9 (**C**) western blot of uric acid transport-related protein expression in kidney tissue; the expression of (**D**) URAT1 and (**E**) GLUT9 proteins. Data are expressed as mean ± S.E.M. (n = 6), * or ^#^
*p* < 0.05, ** or ^##^
*p* < 0.01, *** or ^###^
*p* < 0.001, **** or ^####^
*p* < 0.0001 (^#^ Compared with the NC group and * is compared to M Group). NC was control group. M was HUA model group. AP group was treated with the positive drug allopurinol group. GQ01 group was given live *Pediococcus acidilactici* GQ01. G1PB group was given inactivate *Pediococcus acidilactici* GQ01.

**Figure 6 foods-13-00923-f006:**
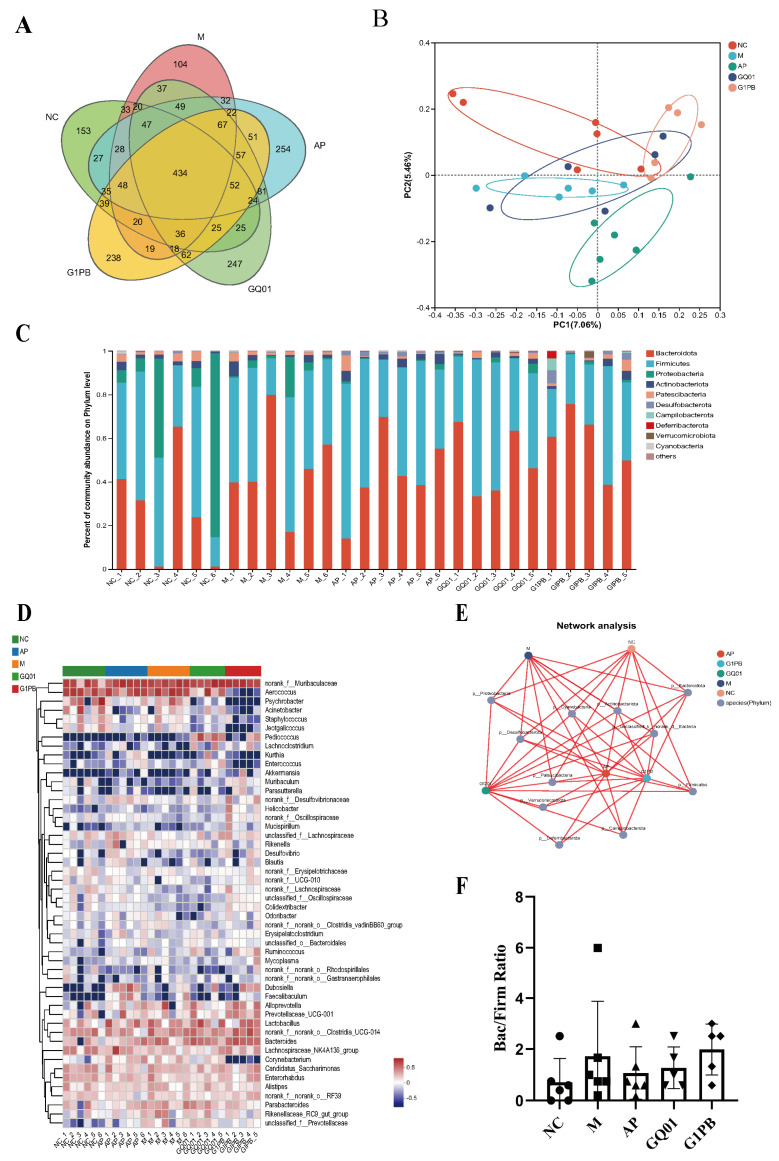
Effects of *P. acidilactici* GQ01 and G1PB postbiotics on gut microbe of mice. (**A**) Venn diagram analysis of OTU overlaps between different groups of microorganisms; (**B**) PCoA principal component analysis; (**C**) effects of GQ01 and G1PB on the composition of gut flora at the portal level in HUA mice; (**D**) effects of GQ01 and G1PB on gut flora composition at the genus level in HUA mice; (**E**) distribution of species and different groups; and (**F**) changes in the Bac/Firm ratio in the different groups. Data are expressed as the mean ± SEM (n = 6). NC was control group. M was HUA model group. AP group was treated with the positive drug allopurinol group. GQ01 group was given live *Pediococcus acidilactici* GQ01. G1PB group was given inactivate *Pediococcus acidilactici* GQ01.

**Figure 7 foods-13-00923-f007:**
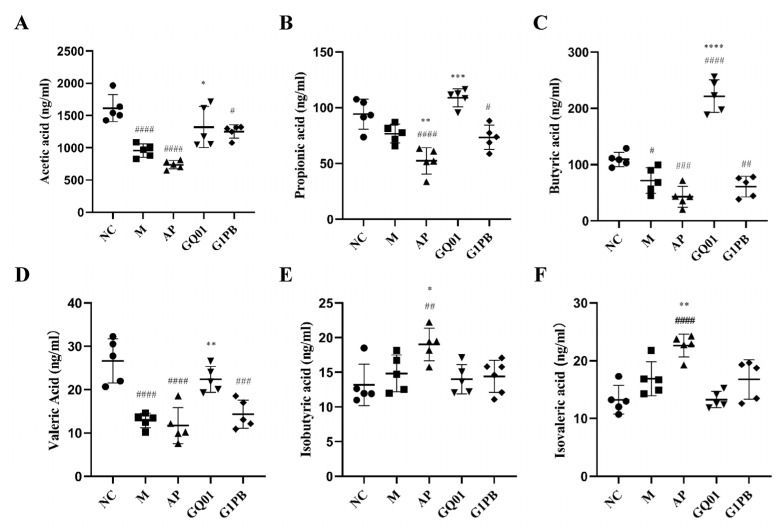
Effect of *P. acidilactici* GQ01 and G1PB on fatty acid content of colonic metabolites in a hyperuricaemia model mouse. (**A**) Acetic acid; (**B**) propionic acid; (**C**) butyric acid; (**D**) valeric acid; (**E**) isobutyric acid; and (**F**) isovaleric acid content. Data are expressed as the mean ± SEM (n = 6). * or ^#^
*p* < 0.05, ** or ^##^
*p* < 0.01, *** or ^###^
*p* < 0.001, **** or ^####^
*p* < 0.0001 (^#^ Compared with the NC group and * is compared to M Group). NC was control group. M was HUA model group. AP group was treated with the positive drug allopurinol group. GQ01 group was given live *Pediococcus acidilactici* GQ01. G1PB group was given inactivate *Pediococcus acidilactici* GQ01.

**Table 1 foods-13-00923-t001:** Target gene primer sequences.

Gene	Forward Primer	Reverse Primer
β-actin	GTGACGTTGACATCCGTAAAGA	GTAACAGTCCGCCTAGAAGCAC
ABCG2	GGCCTGGACAAAGTAGCAGA	GTTGTGGGCTCATCCAGGAA
XOD	ATGACGAGGACAACGGTAGAT	TCATACTTGGAGATCATCACGGT
GLUT9	GATGCTCATTGTGGGACGGTT	GATGCTCATTGTGGGACGGTT

## Data Availability

The original contributions presented in the study are included in the article, further inquiries can be directed to the corresponding author.

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
