# Peer review of "Postbiotic of Pediococcus acidilactici GQ01, a Novel Probiotic Strain Isolated from Natural Fermented Wolfberry, Attenuates Hyperuricaemia in Mice through Modulating Uric Acid Metabolism and Gut Microbiota"

_foods, 2024, doi:10.3390/foods13060923_

Round 1

Reviewer 1 Report

Comments and Suggestions for Authors

The manuscript from Ren et al. describes the isolation of a novel potentially  probiotic isolate obtained from natural fermented wolfberry. The isolate was identified as Pediococcus acidilactici GQ01 and was evaluated in a mice model of hyperuricemia. The probiotic and it’s postbiotic preparation attenuate hyperuricemia in mice through modulating uric acid metabolism and gut microbiota.

The manuscript is well written and organized. The experiments were well designed. But it till Need some adjustments.

Major issues 

The authors should provide more information for classify the isolate as probiotic. What characteristics the isolate has to be classified in this way. 

Minor issues 
- Please attempt to use ‘intestinal microbiota’ instead of ‘’intestinal flora’.

- Other issues are provided in the attached pdf file.

Author Response

Point 1: The authors should provide more information for classify the isolate as probiotic. What characteristics the isolate has to be classified in this way.

Response 1: Thank you for pointing this out. I have re-added the third paragraph in the Introduction of the manuscript, which describes the relationship between gut flora and probiotics, the introduction of the probiotic concept, the role probiotics play in the body, and the types of probiotics. Manuscript line number (74-80)

Point 2: Please attempt to use ‘intestinal microbiota’ instead of ‘intestinal flora’.

Response 2: Thanks to your suggestion, I have revised the whole manuscript and your suggestions have made the article more rigorous.

Point 3:Annotation comment in manuscript PDF】Please consider to provide another title for this subsection such as: Determination of growth kinetic and organic acids production.

Response 3: Thank you for pointing this out. I agree with this comment. I have changed subheading 2.3. Manuscript line number 132

Point 4:Annotation comment in manuscript PDF】Please provide the meaning of each abbreviation for the groups in the figure legend.

Response 4: Thanks to your suggestion. I add the meaning of each group of abbreviations to the legend from Figure 2 to Figure 7. (NC was control group. M was HUA model group. AP was treated with the positive drug allopurinol group. GQ01 was given live Pediococcus acidilactici GQ01. G1PB was given inactivate Pediococcus acidilactici GQ01)

Reviewer 2 Report

Comments and Suggestions for Authors

Lu Ren et al explore the Postbiotic of Pediococcus acidilactici GQ01, a novel probiotic 2 strain isolated from natural fermented wolfberry, attenuate 3 hyperuricemia in mice through modulating uric acid metabolism and gut microbiota, which is a very fascinating and interesting topic, the paper was well written, however I have some minor concerns to be clarified by authors.

1.     What is the relationship between gut microbiota and hyperuricemia? The authors should introduce these lines in the introduction.

2.     It might be beneficial to provide a brief explanation or definition of "postbiotic" for readers who may not be familiar with the term. This would help ensure that the title is accessible to a wider audience and avoids potential confusion.

3.     The mention of the strain being isolated from natural fermented wolfberry adds an intriguing aspect to the research study. However, it could be helpful to briefly explain why this source is significant or relevant. Supporting context on the potential benefits or unique properties associated with strains isolated from natural sources could enhance reader interest.

4.     Put all the plants animal bacterial specie in the Italics see line 118 (Lycium barba- rum)

5.     Why authors did westrenblot in the kidneys only?

6.     Figure abbreviations should be in the legends. Does NC mean negative control or normal control?

7.     Figure should be the consistent font, color, and size. Please compare figure 1 with the other figures, for instance.

Comments on the Quality of English Language

May need english correction moderately 

Author Response

Point 1: What is the relationship between gut microbiota and hyperuricemia? The authors should introduce these lines in the introduction.

Response 1: Thanks to your suggestion, we've added this part of the argument to the manuscript: Intestinal microbiota and its metabolites play an important role in human uric acid metabolism, intestinal flora can participate in purine absorption, synthesis of uricase and affect the intestinal mucosal barrier, and thus directly or indirectly involved in uric acid production and excretion. Manuscript line number 57-60. And in the second paragraph of Introductin, lines 55-73, the relationship between gut microbiota and hyperuricaemia and the influence of each other is outlined.

Point 2: It might be beneficial to provide a brief explanation or definition of "postbiotic" for readers who may not be familiar with the term. This would help ensure that the title is accessible to a wider audience and avoids potential confusion.

Response 2: Thanks to your suggestion, we added lines 82-95 of the manuscript about when postbiotics were first introduced and the concept, and the consensus definition of postbiotics published by the International Society for the Science of Probiotics and Prebiotics (ISAPP) in May 2021, and compared the features and benefits of postbiotics compared to live probiotics.

Point 3: The mention of the strain being isolated from natural fermented wolfberry adds an intriguing aspect to the research study. However, it could be helpful to briefly explain why this source is significant or relevant. Supporting context on the potential benefits or unique properties associated with strains isolated from natural sources could enhance reader interest.

Response 3: Thanks to your suggestion. Therefore, we introduced wolfberry (Lycium barbarum L.), a medicinal food with a long history, in the fourth paragraph of Introduction. Probiotic fermented fruit, vegetable and dairy products exist in large quantities in the market, and the fermented products not only have the original taste and nutrients, but also produce the unique flavour and nutrients of fermentation, but there are very few products of probiotic fermented wolfberry. Therefore, it is very important to screen functional probiotics suitable for fermentation and metabolic transformation of wolfberry. The natural wolfberry fermented products (Jiaosu) is just a good source of isolation for such probiotics (Manuscript line number 96-107).

Point 4: Put all the plants animal bacterial specie in the Italics see line 118 (Lycium barbarum)

Response 4: Thanks to your suggestion, I have revised the whole manuscript.

Point 5: Why authors did westrenblot in the kidneys only?

Response 5: Thanks to your suggestion. Because changes in XOD at the mRNA level in liver have been demonstrated in Fig. 3C, and Fig. 3B shows the results of the test for XOD enzyme viability, enzyme activity is positively correlated with protein level, and enzyme activity not only responds to protein level, but also to the viability of the XOD enzyme, the proteins of XOD in liver have been included in the test for enzyme viability. However, the two proteins tested in the kidney, as transporter carriers, could not be detected by the determination of the viability, so westrenblot were done in the kidneys.

Point 6: Figure abbreviations should be in the legends. Does NC mean negative control or normal control?

Response 6: Thanks to your suggestion, I have revised the whole manuscript. NC is the normal control group.

Point 7: Figure should be the consistent font, color, and size. Please compare figure 1 with the other figures, for instance.

Response 7: Thanks to your suggestion, I've corrected the image size fonts.

Response to Comments on the Quality of English Language

Point 1: May need english correction moderately.

Response 1: Thank you for your valuable and thoughtful comments. We have carefully checked and improved the English writing in the revised manuscript.
